# A Systematic Review of Positional Plagiocephaly Prevention Methods for Patients in Development

Alessio Danilo Inchingolo [1,†], Angelo Michele Inchingolo [1,†], Fabio Piras [1,†], Giuseppina Malcangi [1,†], Assunta Patano [1], Chiara Di Pede [1], Anna Netti [1], Anna Maria Ciocia [1], Alberto Corriero [2], Alexandra Semjonova [1], Daniela Azzollini [1], Elisabetta De Ruvo [1], Fabio Viapiano [1], Irene Ferrara [1], Giulia Palmieri [1], Merigrazia Campanelli [1], Antonio Mancini [1], Nicole De Leonardis [1], Pasquale Avantario [1], Silvio Buongiorno [1], Maria Celeste Fatone [3], Stefania Costa [4], Valentina Montenegro [1], Gianluca Martino Tartaglia [5], Biagio Rapone [1,*], Ioana Roxana Bordea [6], Antonio Scarano [7], Felice Lorusso [7], Andrea Palermo [8], Daniela Di Venere [1,‡], Francesco Inchingolo [1,*,‡] and Gianna Dipalma [1,‡]

1 Department of Interdisciplinary Medicine, University of Bari "Aldo Moro", 70124 Bari, Italy
2 Department of Interdisciplinary Medicine, Intensive Care Unit Section, Aldo Moro University, 70121 Bari, Italy
3 PTA Trani-ASL BT, Viale Padre Pio, 76125 Trani, Italy
4 Department of Biomedical and Dental Sciences and Morphofunctional Imaging, Section of Orthodontics, School of Dentistry, University of Messina, 98125 Messina, Italy
5 UOC Maxillo-Facial Surgery and Dentistry, Department of Biomedical, Surgical and Dental Sciences, School of Dentistry, Fondazione IRCCS Ca' Granda, Ospedale Maggiore Policlinico, University of Milan, 20100 Milan, Italy
6 Department of Oral Rehabilitation, Faculty of Dentistry, Iuliu Hațieganu University of Medicine and Pharmacy, 400012 Cluj-Napoca, Romania
7 Department of Innovative Technologies in Medicine and Dentistry, University of Chieti-Pescara, 66100 Chieti, Italy
8 Implant Dentistry College of Medicine and Dentistry Birmingham, University of Birmingham, Birmingham B4 6BN, UK
* Correspondence: biagiorapone79@gmail.com (B.R.); francesco.inchingolo@uniba.it (F.I.); Tel.: +39-347-761-9817 (B.R.); +39-331-211-1104 (F.I.)
† These authors contributed equally to this work as first authors.
‡ These authors contributed equally to this work as the last authors.

**Abstract:** Positional plagiocephaly is an asymmetrical skull deformation caused by various factors. Although it is not responsible for abnormal brain development in infants and is not related to the onset of neurophysiological problems, it is critical to prevent skull deformity to avoid aesthetic and functional consequences. The purpose of the study is to investigate the relevance of preventive procedures to the onset of positional plagiocephaly, such as the use of passive mattresses, which is primarily correlated with the need for newborns and infants to sleep and rest in proper posture. PubMed, Web of Science, Google Scholar, Scopus, Cochrane Library, and Embase were searched for papers that matched our topic, dating from January 2012 to 22 October 2022, with an English language restriction, using the following Boolean keywords: ("positional plagiocephaly" AND "prevention"). A total of 11 papers were included as relevant papers matching the purpose of our investigation. According to the research results, inadequate vitamin D and folic acid intake during pregnancy may increase the risk of skull deformation. Furthermore, babies should sleep on their backs and spend at least 30 min in tummy time. Using a passive sleep curve mattress has several advantages such as low cost, easy handling, no compliance system, and a marked improvement in head shape, allowing harmonious skull growth guided by normal brain expansion.

**Keywords:** positional plagiocephaly; rigid mattress; flat head; semi-rigid mattress; infant well-being; infant cranial development; cranial deformation

## 1. Introduction

The term Plagiocephaly is Greek (plagios = obliqua, kefale' = head) and stands for all alterations in the shape of the head, which can originate from multiple causes [1].

Plagiocephaly predominantly affects the infant [2].

Two forms of plagiocephaly can be distinguished: that due to premature fusion of cranial sutures (synostotic plagiocephaly) and that generated by external pressures affecting the pre- and postnatal growth orientation of the skull (deformational and/or positional plagiocephaly) [1,3–5].

Pediatricians are obligated to examine children with plagiocephaly, which can never be surpassed by impulsive and erroneous assumptions or therapy [6]. Prompt clinical experience is the cornerstone of cranial deformity treatment. In syndromic associations and skeletal dysplasia textbooks and articles, many syndromic entities fulfill the search criteria [7]. Characteristics of plagiocephaly, in addition to non-instantaneous and perplexing versions of sclerosing diseases of the skeleton, craniotubular abnormalities include a lengthy list [8–10]. The order of precedence management should consist of the following: pediatricians and doctors must first conduct a thorough examination of the youngster, which should involve the parents; clinical observations are required regardless of severity for any uncommon craniofacial/musculoskeletal anomalies; secondly, they must conduct a clinical evaluation of the siblings and information collection for the remainder of the family; thirdly, doctors must rule out every kind of aberrant craniofacial shape and their causes and related syndromic connections [11].

Positional Plagiocephaly (PP) also can involve one or both sides of the occiput [12–15] (Figure 1).

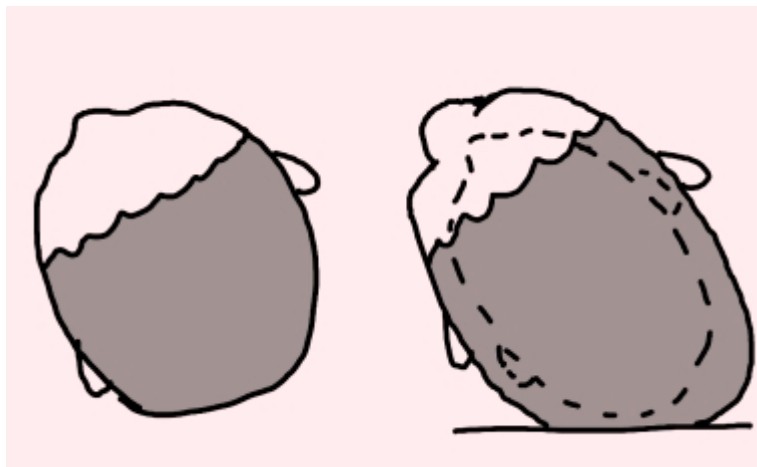

**Figure 1.** The pathogenetic mechanism of skull deformation.

Cranial sutures are classified as synarthrosis, a fixed joint type that allows no movement [16]. In the fetal skull, these sutures are non-rigid and permit some mobility during birth [17].

At birth, the skull's bones can overlap, a process known as *molding*. Physiologically, the skull adapts to the progressive development of the brain, increasing in volume during the infant's growth [18].

PP does not cause abnormal brain development or neuropsychological problems [19–21].

The infant's plastic, malleable skull can have intentional, pathological, or paraphysiological deformities [22]. The latter occurrence usually happens during delivery, at the passage of the vaginal canal, and is usually reversible with the recovery of shape after a few weeks of life [23].

Neonatal PP occurs in those infants who show a preferential side of head support in the first few months of life [24,25]. Male sex, twin pregnancies, preterm deliveries, use of forceps in assisted delivery, and torticollis are thought to be risk factors [26–28].

Once posterior flattening has occurred, spontaneous correction of the skull does not occur because the external pressure of the bearing surface acts more incisively and continuously on the flattened part, despite the later, more frequent change in the position of the child's head [13,29,30].

Since 1992, when the American Academy of Pediatrics declared the prone position responsible for sudden infant deaths and recommended a supine position (Back to Sleep campaign), the percentage of PP and DB has increased from 0.3% to 48% [31,32].

Interventions by pediatric plastic surgeons and neurosurgeons in the treatment of PP have markedly increased, because parents are concerned about the consequences of PP on the child's appearance, development, and future relationships [32–35].

The functional consequences of plagiocephaly are postural and musculoskeletal disorders, sensory disorders, and impact on neurological development [36].

Therapy, which involves different protocols, depends on the level of asymmetry, estimated according to Argenta's classification [37–39]. Argenta's classification recognizes five levels of PP severity (Figure 2) [40].

(1) Type I PP (PPI), the mildest form, in which asymmetry is present only in the back of the skull and whose depression may vary. There are no positional asymmetries at the ears level, deformities in the frontal region, or vertical changes (elongations) of the face.

(2) Type II PP (PPII), a more advanced form of PP, in which various levels of posterior cranial asymmetry with involvement of the skull base and temporal fossa are recognized. The ear of the affected side is located forward and/or downward (visible when looking at the infant's skull from above). The frontal squama (symmetrical forehead) is not involved. There is no facial asymmetry or vertical (compressive) changes in the skull.

(3) Type III PP (PPIII), characterized by the typical parallelogram shape of the skull. There is posterior cranial deformation, changing of the ear from the affected part, and involvement of the frontal squama ipsilateral to the depression with a prominence of the same (visible from above). The face appears symmetrical.

(4) Type IV PP (PPIV), characterized by posterior cranial deformity associated with facial asymmetry, frontal and ear swiveling ipsilateral to the cranial depression. Hyperplasia of the zygomatic process (less frequent) and displacement of cheek adipose tissue due to the progression of cranial deformation result in facial asymmetry.

(5) Type V PP (PPV) in which, in addition to posterior cranial deformity associated with facial asymmetry, frontal and ear slippage ipsilateral to cranial depression, in the occipitoparietal area, there is altered vertical growth of the skull, as well as a protrusion in the temporal area [39].

Healthy infants progressively change, from three to nine weeks, their tendency to position the head laterally to a degree symmetrical with the position along the midline at nine weeks. A correlation was found between the duration and strength of head orientation and the occurrence of PP [41,42].

Infants with severe deformities have been seen to begin HT with an asymmetry of the cranium (AC) that improves, albeit slowly, with its use, and the greatest improvements are achieved in the first two months of treatment [43–45].

The use of orthotic helmet has many disadvantages: malodorous perspiration, nonreimbursable cost, the social stigma of the helmet, poor helmet fit, skin injury over pressure points, skin reaction to helmet lining materials, poor fit with severe brachycephaly, and hair loss [46,47].

According to data from the AAP, the rates of plagiocephaly have increased exponentially (from 0.3% to 45%). Our research intends to draw attention to the best options available in the literature for a clinical problem that, if considered correctly, could be prevented with simple and inexpensive therapy devices.

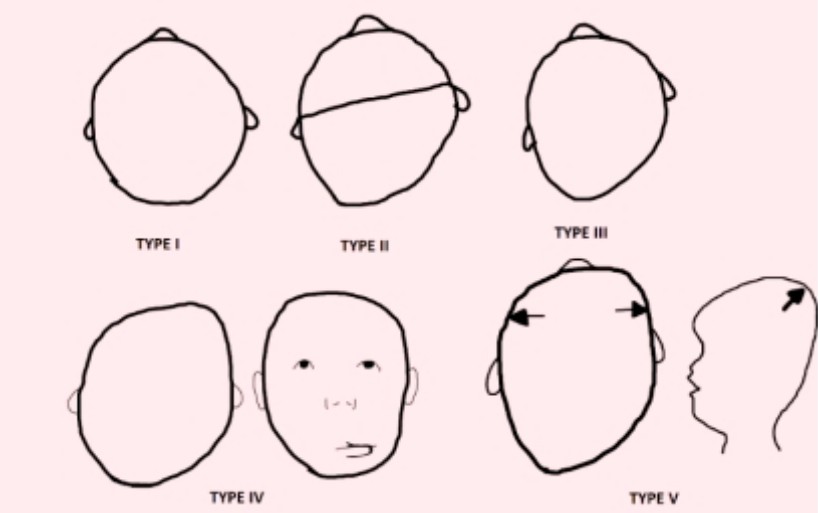

**Figure 2.** Classification into 5 types of plagiocephaly according to Argenta.

## 2. Materials and Methods

### 2.1. Search Processing

The present systematic review was conducted following the PRISMA and International Prospective Register of Systematic Review Registry protocols (full ID: CRD42022341814). From January 2012 to 22 October 2022, PubMed, Web of Science, Google Scholar, Scopus, Cochrane Library, and ScienceDirect were searched for works that matched our topic, with an English language restriction. Hence, the following Boolean keywords (Table 1) were used: ("positional plagiocephaly" AND "prevention*").

**Table 1.** Database search indicators.

| | |
|---|---|
| Articles screening strategy | KEYWORDS: A: "positional plagiocephaly"; B: "prevention"; Boolean Indicators: "A" AND "B" Timespan: from January 2012 up to October 2022. Language: only ENGLISH Electronic Databases: PubMed, Web of Science, Google Scholar, Scopus, Cochrane Library, ScienceDirect |

### 2.2. Inclusion Criteria

Reviewers worked in pairs to analyze all appropriate studies that met the following inclusion criteria: (1) babies and children aged 0 to 2 years; (2) open access papers that other researchers may view without paying a fee; (3) research that investigated the preventative interventions against the development of PP, such as the use of a hard-surfaced mattress, lateralization of the infant during sleep, and tummy time activities.

### 2.3. Data Processing

The quality of the included studies was appraised by two independent reviewers (F.P. and A.C.) using predetermined criteria such as selection criteria, methods of outcome evaluation, and data analysis. The inclusion criteria were used to identify any potentially published articles with full texts. Any disagreements were resolved by discussion or collaboration with a third researcher (F.I.).

## 3. Results

### Characteristics of Included Articles

A total of 530 articles were discovered, utilizing six databases, including PubMed (110), Web of Science (37), Google Scholar (138), Scopus (187), Cochrane Library (7), and ScienceDirect (123), generating 416 results after duplicates were removed (114). The ti-

tle and abstract analysis resulted in the elimination of 247 publications. The remaining 169 records were retrieved, obtaining 71 reports that the authors assessed for eligibility. Because they were off topic, 60 publications were rejected from the discussion. In total, 11 papers were included in the review for qualitative analysis (Figure 3).

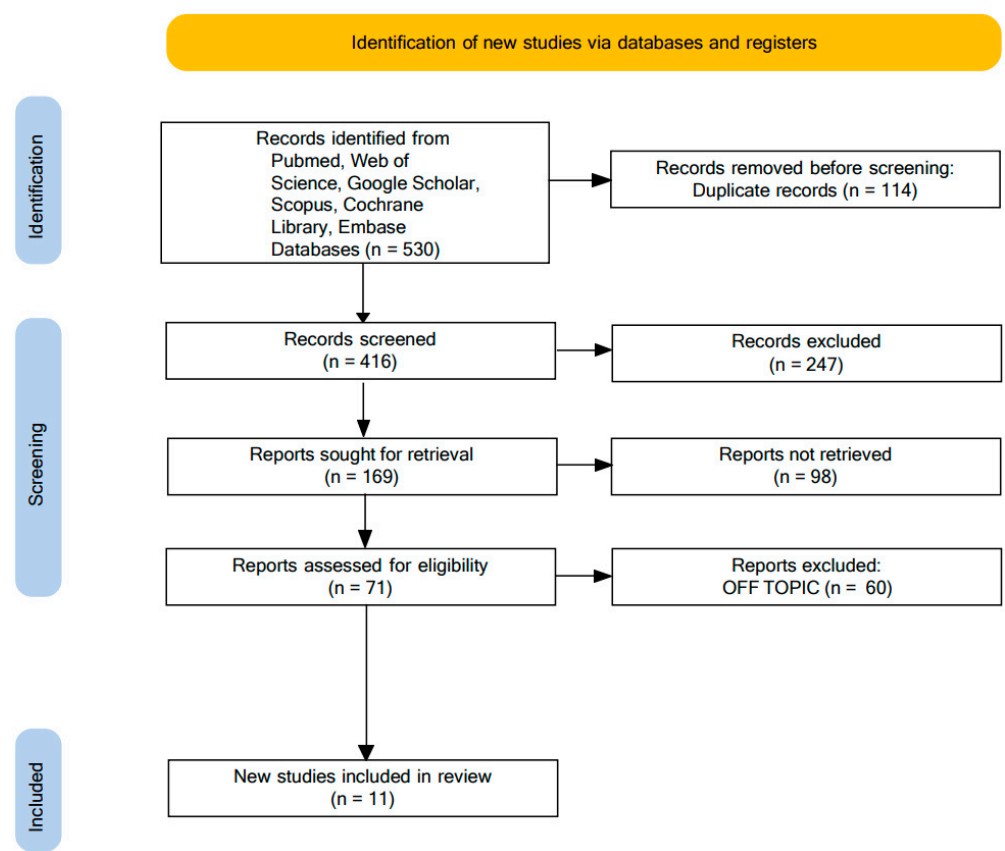

**Figure 3.** PRISMA flowchart diagram of the inclusion process.

## 4. Discussion

### 4.1. Cranial Conformation-Related Positional Plagiocephaly

In the studies under review, progressive exposure to the supine pose was assessed in several ways, including the extent of time expended in the pose each day and time spent in the side-line, prone, sleeping, and feeding positions (Table 2).

**Table 2.** Studies about cranial conformation related to PP.

| Ref | Authors (Year) | Type of the Study/Days | Aim of the Study | Materials | Results |
|-----|----------------|------------------------|------------------|-----------|---------|
| [48] | Miyabayashi et al., 2022 | Observational study | 3D scanner evaluation of values for cranial morphological features in 1-month-old Japanese infants, to have the prevalence of PP. | In 153 healthy infants, a 3D scanner and image analysis software was used to evaluate cranial form. | The CVAI (Cranial Vault Asymmetry Index) >3.5% or ≥10% were diagnosed as deformational (64.7%) or severe PP (6.6%). |

| Ref | Authors (Year) | Type of the Study/Days | Aim of the Study | Materials | Results |
|---|---|---|---|---|---|
| [49] | Wang Yang et al., 2019 | Multicenter study | Evaluation of PP in premature infants to have an early diagnosis. | In 530 preterm infants, the head shape was evaluated by a simple manual method based on Wilbrand's standardized scheme. | PP was 3.0%, and those of right and left plagiocephaly were 69.4 and 30.6%, respectively. The PP was highest in newborns with a gestational period < 32 weeks. |
| [50] | Kajita et al., 2017 | Case report | Evaluation of ULS with typical cranium characteristics of PP. | 10-month girl; 3-year-old boy. | Collaboration with craniofacial surgeons is fundamental in the case of infant cranial deformity to avoid simply diagnosing PP and missing craniosynostosis. |
| [41] | Amy Y.F. Leung et al., 2016 | Prospective study | Evaluation of skull orientation and how PP relates to it in full-term newborns with no congenital problems. | 94 newborns were observed at 3, 6, and 9 weeks to control head orientation, including head orientation length, strength, and turning latency. At three weeks, the direction of the head on one side duration was most prevalent (right: 40, left: 41, midline: 19). | After the side was taken into consideration, head orientation force was comparable throughout the three ages, although from 3 to 9 weeks, the vigor of head orientation to the left declined. |
| [14] | Cabrera-Martos et al., 2013 | Prospective clinical trial | According to their unique clinical profile, the effectiveness of a conservative strategy in plagiocephaly-affected infants is evaluated. | 104 newborns with nonsynostotic plagiocephaly accompanied or not by congenital or positional torticollis. | The suggested physiotherapy regimen can successfully treat Plagiocephaly. |
| [51] | Ifflaender et al., 2013 | observational study | Prevalence of symmetrical and AC abnormalities and to pinpoint potential risk factors, data on the head shape of preterm newborns at TEA (Term-Equivalent Age) are evaluated. | 195 newborn scans were gained using a 3D laser shape digitizer. | At TEA, preterm newborns had a significant frequency of symmetrical and AC abnormalities. |

Plagiocephaly has different forms. The other form of plagiocephaly, PP, is rare in newborns, but affects 19.7% of healthy 4-month-old babies [52].

### 4.2. Prevention of Positional Plagiocephaly and Risk Factors

Plagiocephaly is characterized by changes in the shape of the skull, and if not diagnosed and treated early, it can be a lifelong condition [53]. Therefore, identifying risk factors is necessary to adequately inform the parents and guardians of children.

Scientific literature has investigated the correlation between positional deformity of the skull, and vitamin D. Vitamin D is essential for bone mineralization; therefore, deficiency during childhood can cause the bones of the head to remain more malleable and deform as a result of pressure [54].

The study by Weernink et al. [55] investigates the correlation between positional deformation of the skull in children between the ages of 2 and 4 months and the intake of Vitamin D supplements (10 µg per day) during pregnancy and the first months of the child's life (Table 3).

**Table 3.** Risk factors of PP.

| Ref | Authors (Year) | Type of the Study/Days | Aim of the Study | Materials | Results |
|---|---|---|---|---|---|
| [55] | Weernink et al., 2014 | Observational case–control study | Analyzing the relationship between intake of vitamin D supplements during the third trimester of pregnancy (mother) and during the newborn period (child) and the emergence of positional skull deformation in infants aged 2 to 4 months. | 548 matched controls and 275 babies with positional cranial deformation aged 2–4 months were compared. | An increased risk of positional deformation of the skull is linked to the pregnant woman's inadequate vitamin D consumption. |
| [56] | Michels et al., 2012 | Retrospective case–control study | Evaluation of the connection between elevated folic acid intake and PP. | Participants in the study were 94 mothers (PP group: children with PP) and 94 mothers (CO group: children without PP). Some subjects were excluded from the study, so the PP group consisted of 75 mothers and the CO group of 54. | 20% of PP patients took the recommended double dose of folic acid, while only 6% of the CO group % did the same. |
| [57] | Nuysink et al., 2012 | Retrospective longitudinal study | Examine the frequency of an idiopathic asymmetry in 192 newborns (gestational age ≤ 32.0 weeks) at TEA and 6 months CA (Corrected age). | 192 infants born ≤ 32.0 weeks of gestation | At TEA, positioning preferences for the head were present in 44.8% of cases; in 10.4% of PP cases, and 13 percent of cases at 6 months CA. In preventing the onset of a PP, special care must be taken, given the high frequency of a positional preference in children delivered preterm at TEA. |

**Table 3.** *Cont.*

| Ref | Authors (Year) | Type of the Study/Days | Aim of the Study | Materials | Results |
|-----|----------------|------------------------|------------------|-----------|---------|
| [26] | Mawji et al., 2014 | Prospective cohort study | Identify possible causes of PP in newborns between the ages of seven and twelve weeks. | 384 healthy-term babies aged 7 to 12 weeks | It was calculated that 46.6% of children had PP. The identified risk factors are a right-sided head positional preference, supine in sleep, part positioned by vacuum/forceps, and male sex. |

An increased skull malformation risk has been linked to inadequate vitamin D consumption in early infancy and during the final trimester of pregnancy [55].

Furthermore, the present study [55] found that infant formula consumption after birth was connected to a higher risk of cranial deformity, despite being a source of vitamin D. This seems to be because of the one-sided position during bottle feeding [55].

Michels et al. [56] evaluated the relationship between high folic acid intake and PP.

To prevent neural tube defects, a daily dose of 400 μg of folic acid is recommended in the Netherlands from the fourth week before conception until the eighth week of pregnancy [58].

As a result, mothers of children with PP tend to consume too much folic acid each day while pregnant. Further research should be conducted to understand how folic acid might affect bone development (Table 3) [56].

Variables that could lead to positional cranial deformity have been identified in the literature, including multiple-birth pregnancy, premature gestation, prolonged labor, breech presentation and assisted delivery (vacuum/forceps) [37,59–61], positional preference, lack of variation in head positioning when sleeping in the first six weeks of life, and position during bottle feeding [60–63].

Nuysink et al. [57] investigated the correlation between the positional preference and PP in preterm babies. The study analyzed 192 infants born ≤ 32.0 weeks of gestation.

A positional preference was present in 44.8% of newborns in our study who were term-equivalent aged (TEA), compared to the 13–20% found in full-term born infants [52,61,64], but only 10.4% had a PP. At six months CA, PP is observed in 13% of the infants (Table 3).

A plagiocephaly is three times more likely to occur in infants at six months of age with a postural preference for TEA than in those without [57].

The positioning of the equipment on the right side of the incubators may have contributed to establishing a positional preference. Furthermore, the low maturity of the system in preterm babies leads to asymmetrical postures given by different muscle tones [65–69].

Five risk factors were linked to plagiocephaly: supine sleeping position, sex, type of delivery, preference for the right head position, and preference for the left head position.

The outcomes revealed that infants sleeping supine were approximately 2.7 times more likely to develop PP than infants who were not placed supine. Supine positioning is suggested to decrease the incidence of SIDS, so it is not considered a modifiable risk factor [26].

The type of delivery is a plausible risk factor because there may be compression of the infant's skull. Unassisted vaginal deliveries were compared with assisted (forceps and vacuum) and cesarean deliveries.

### 4.3. Positional Plagiocephaly and Mattress

The current management of PP involves the use of an orthotic mattress. This could improve the redistribution of pressure on a surface where the infant's skull could normally grow, guided by the brain's expansion.

From the early 1990s, the incidence of PP rapidly increased with the "back to sleep" campaign, which promoted the supine position to reduce the risk of sudden infant death [29].

To prevent plagiocephaly, the AAP counsels that newborns should avoid the prone position to sleep, sleep on the back, and stay at a minimum of 30 min in tummy time [70].

An important point to enhance is that the risk of SIDS is 18-fold greater in newborns that sporadically sleep in a prone position [70].

Compared to other protocols based on repositioning advice or helmet orthotics, a sleep curve mattress does not require the child's compliance and is less expensive [71,72].

A study by Sillifant et al. describes the changes in head shape using a passive orthotic mattress to treat a group of thirty infants with PP (Table 4) [73].

**Table 4.** Sleep curve mattress for the management of PP.

| Ref | Authors (Year) | Type of the Study/Days | Aim of the Study | Materials | Results |
|---|---|---|---|---|---|
| [73] | Sillifant et al., 2014 | Clinical study | The objective is a comparison of cranial deformity of infants affected by PP before and after the adoption of a Sleepcurve mattress. | 30 patients with a mean age of five months were included. The grade of asymmetry was assessed with CVA and was classified into three ranges: clinically insignificant (CVA 6 mm), moderate (CVA 6–12 mm), and high (CVA > 12 mm). | The improvement of cranial symmetry was statistically relevant (*p* value of 0.001). The average asymmetry was 5 mm, compared to the 16 mm at the start. The symmetry enhancement was 11 mm. |

The mattress is like a passive orthotic device, creating a concavity where the head of the infant can fit. It is intended to reduce the effect of repetitive sleeping position that creates a flat surface.

The use of an orthotic mattress in the management of PP showed a reasonable improvement in head form compared to other treatment protocols. Most patients with a severe grade of asymmetry showed minimal deformity at the end of the research [73].

Mobility is one of the most significant accomplishments of infancy, since it allows the infant to become independent and gradually explore new environments and objects. Rolling over from supine to prone is one type of mobility that develops in a child [74].

When a baby turns from supine to prone at the age of six months, this change in body posture is required, and it also marks the beginning of sitting without support [75].

Tummy time is a crucial kind of physical activity for non-mobile infants, according to national and international guidelines [76–78].

As infants grow, tummy time can also help them become more stable in weight-bearing positions such as sitting and lying prone on their hands and knees [79].

Infants should be placed for sleep in the supine position (entirely on the back) by every caregiver until they are one year old to lower the risk of SIDS. Side sleeping is not recommended or safe [31,70].

A supine position does not increase choking and aspiration risks. Only infants with specific upper airway abnormalities, such as laryngeal clefts, should be considered to be placed in the prone position while they sleep, since the danger of dying from gastric reflux illness may be greater than the risk of SIDS in these infants [31,80].

A semi-rigid mattress could be essential for ensuring correct rest and respiration [81–83].

The prevalence of SIDS has decreased where supine sleeping has been promoted, although the cause is unknown. One potential cause is the obstruction of the upper airways during sleep. The pharynx has been linked to bouts of apnea based on changes in pharyngeal pressure [84].

Skatvedt et al. demonstrated that compared to the supine head rotated and prone postures, the peak negative inspiratory pressures (PES) measured from the esophagus were considerably lower in the supine head straight position [84].

Due to pathophysiological and anatomical reasons, newborns are more vulnerable to respiratory muscle weakness, muscle exhaustion, and respiratory failure [85].

TAA has been proven to be less severe in the prone position than in the supine position [86].

The surface allows a harmonious cranial shape and avoids possible plagiocephaly. It helps to preserve healthy and free sealing of the posterior fontanel, which closes between 6 weeks and 3 months of the baby's life [87,88].

Although there is no evidence in the literature, from an evaluation of products on the market (Figure 4), according to our experience, the Welcome Pad® mattress presents more features to enable the comfort, hygiene, and safety of the child. It is an evolution of a device studied, by professionals from the departments of Neonatology and Neonatal Intensive Care at the Ospedale Maggiore in Bologna and the Infant Rehabilitation Medicine at Bellaria, of which no data have been published [89]. It has been patented as ornamental and filed on 25 July 2017 with n° 004119121 at the community level and registered with an official certificate. The registration number is 004119121 in 2019 (COMMUNITY MODEL).

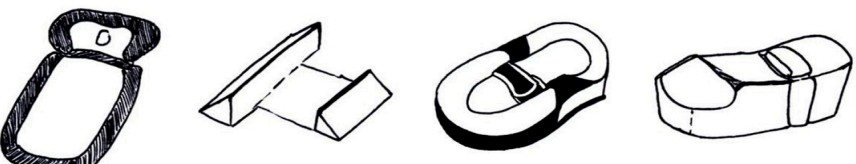

**Figure 4.** Some orthopedic mattresses are on the market.

Welcome Pad® is breathable, allowing the newborn and infant to regulate their temperature physiologically. The thermoregulating effect allows for avoidance of heat stroke and overheating. Its natural fiber cover is plastic free. It is also completely removable and machine washable to avoid the risk of allergies or skin hypersensitivity. Furthermore, its semi-rigid structure prevents the risk of suffocation and ensures maximum comfort (Figures 5 and 6).

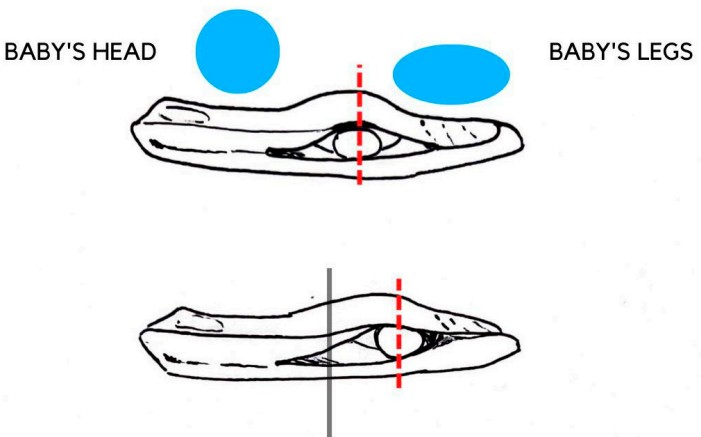

**Figure 5.** Anatomical cylinder of Welcome Pad® Baby Wellness Foudation. Red line: Indicates the position of the pillow referring to an approximate middle line (grey line). The head and legs of the baby rest with the pillow in the middle to assure a correct and safe position.

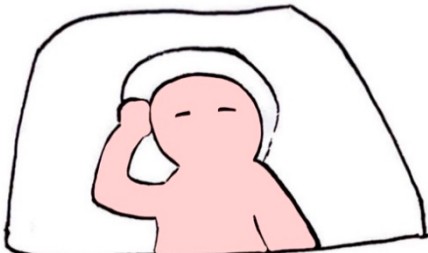

**Figure 6.** Comfortable shape for Newburn's head.

*4.4. Limitations of the Study*

This systematic review discusses the treatment of positional plagiocephaly in detail; unfortunately, regarding mattress therapy, there is currently insufficient scientific literature, although the device appears to be very high performing. There is a lack of clinical trials regarding these devices in the literature due to their recent introduction to the market. Future scientific research involving larger patient samples would be needed to fully understand the mattress pad's effects on positional plagiocephaly.

**5. Conclusions**

Positional plagiocephaly is an alteration in the skull's morphology that develops in infants due to the application of repetitive and deforming forces at the level of the cranial vault. Although the resulting deformities do not represent a short-term health risk for the young patient, they can lead to organic problems such as otolaryngological dysfunction or impaired neurological development. With this systematic review, we want to shed light on the risk factors responsible and the strategies that can be applied to avoid the occurrence of PP, establish a guideline for parents and guardians, and protect the infant from the condition and its possible consequences. In conclusion, it is shown that adequate intake of vitamin D and folic acid during the gestation period, feeding with breast milk without a bottle, and using mattresses prevent the risk of cranial deformations in the infant. The mattress is an essential part of an infant's bedding. The infant spends almost 18 h a day on his mattress during his first few months. This feature makes the mattress an exciting and potentially effective device in preventing the onset of positional plagiocephaly. Compared to other protocols, passive orthotic mattresses have numerous advantages, including low cost, ease of use, and lack of compliance of the child, inducing a significant improvement in head shape. Although extremely promising, these newly introduced devices will require additional research to prove their efficacy and compare their benefits to traditional prevention treatments.

**Author Contributions:** Conceptualization, A.D.I., A.M.I., G.M., F.P., F.L., B.R., A.P. (Assunta Patano), A.P. (Andrea Palermo) and C.D.P.; methodology, A.M.C., A.C., A.S. (Alexandra Semjonova), G.P., D.A., E.D.R., F.V., I.F., M.C. and A.N.; software, N.D.L., A.M., P.A., S.B., S.C., V.M., F.I. and G.D.; validation, B.R., F.L., I.R.B., A.S. (Antonio Scarano), D.D.V., M.C.F., F.V., V.M., F.I. and G.D.; formal analysis, A.D.I., F.P., A.P. (Assunta Patano), A.P. (Andrea Palermo), A.C., I.F., A.M.C., G.M.T., B.R., D.D.V., F.I. and G.D.; resources, A.D.I., A.M.I., G.M., C.D.P., A.N., D.A., F.V., I.F., M.C. and S.C.; data curation, G.M., N.D.L., A.M., P.A., I.R.B., F.L., F.I. and G.D.; writing—original draft preparation, A.D.I., A.M.I., E.D.R., S.B., F.V., I.F., A.C., B.R., F.P., D.D.V., A.N., D.A., F.I. and G.D.; writing—review and editing, A.P. (Assunta Patano), A.P. (Andrea Palermo), A.C., A.M.C., M.C., N.D.L., M.C.F., G.P., I.R.B. and B.R.; visualization, G.P., C.D.P., B.R., F.L., A.P. (Assunta Patano), A.P. (Andrea Palermo), A.M.C., V.M. and I.R.B.; supervision, A.D.I., A.M.I., F.P., G.M., D.D.V., F.I. and G.D.; project administration, A.S. (Antonio Scarano), I.R.B., B.R., G.M.T., F.L., F.I. and G.D. All authors have read and agreed to the published version of the manuscript.

**Funding:** This research received no external funding.

**Institutional Review Board Statement:** Not applicable.

**Informed Consent Statement:** Not applicable.

**Data Availability Statement:** Not applicable.

**Conflicts of Interest:** The authors declare no conflict of interest.

**Abbreviations**

| | |
|---|---|
| 3D | Three-Dimensional. |
| AAP | American Academy of Pediatrics. |
| AC | Asymmetry Of Cranium. |
| CA | Corrected Age. |
| CC | Cranial Circumference. |
| CI | Cephalic Index. |
| Crl | Cranial Length. |
| Crv | Cranial Volume. |
| Crw | Cranial Width. |
| CVA | Cranial Vault Asymmetry. |
| CVAI | Cranial Vault Asymmetry Index. |
| DB | Deformative Brachycephaly. |
| HT | Helmet Therapy. |
| LWR | Cranial Length–Width Ratio. |
| PP | Positional Plagiocephaly. |
| PPI | Positional Plagiocephaly Type I. |
| PPII | Positional Plagiocephaly Type II. |
| PPIII | Positional Plagiocephaly Type III. |
| PPIV | Positional Plagiocephaly Type IV. |
| PPV | Positional Plagiocephaly Type V. |
| PT | Positioning Therapy. |
| SE | Sellion. |
| SIDS | Sudden Infant Death Syndrome. |
| TAA | Thoraco-abdominal Asynchrony. |
| TEA | Term-Equivalent Age. |
| TR | Tragus Landmark. |
| ULS | Unilateral Lambdoid Synostosis. |

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
