# Peer review of "A Systematic Review of Positional Plagiocephaly Prevention Methods for Patients in Development"

_applsci, doi:10.3390/app122111172_

Round 1

Reviewer 1 Report (Previous Reviewer 4)

Grammartical error in some places

Author Response

It has been reviewed and corrected.

Reviewer 2 Report (Previous Reviewer 5)

The additions are fine 

Author Response

Thank you for your comments.

This manuscript is a resubmission of an earlier submission. The following is a list of the peer review reports and author responses from that submission.

Round 1

Reviewer 1 Report

I have read this paper with interest, and value the topic, but I do still have concerns and comments on the current text version

The abstract and title suggests a systematic search on preventive practices for positional plagiocephaly, but how can helmets be part of a preventive strategy ? the conclusions reads different on the aims.

Babies should not avoid sleeping in the prone position, but should sleep on the back (otherwise, side position remains unclear for parents and careproviders).

Plagiocephaly predominantly affects the infant, not the newborn (infant, one month until one year, newborn first 4 weeks)

I do not really understand the risk of bias measurement, as this text section needs a reference. Were the papers assessed on quality, or only on bias risk ? If not, why not, and is this not a shortage ? Have citations or references be screened (‘snowball’). Related to the risk bias measurement, figure 4 is suggested, but is not in the paper.

It looks like the discussion is a mix of results and discussion

What is the relevance of 4.1. as these studies read as observational, not so much ‘preventive’ ?

The link with folic acid is an association, so not necessary (and quite likely) not causal, but association ?

What is the relevance of 4.3 if your studies has its focus ‘only’ on prevention, as this describes interventional studies in diagnosed infants (‘therapies’)

The 4.4. is disproportional large, and seems to be supported by one study ? and mix of results and discussion.

In conclusion, I value the effort made, but the authors have to extensively reconsider their aims, and the subsequent text structure.

Author Response

I have read this paper with interest, and value the topic, but I do still have concerns and comments on the current text version

  1. The abstract and title suggests a systematic search on preventive practices for positional plagiocephaly, but how can helmets be part of a preventive strategy ? the conclusions reads different on the aims.
  2. Babies should not avoid sleeping in the prone position, but should sleep on the back (otherwise, side position remains unclear for parents and careproviders).

It has been corrected

  1. Plagiocephaly predominantly affects the infant, not the newborn (infant, one month until one year, newborn first 4 weeks)

It has been corrected

  1. I do not really understand the risk of bias measurement, as this text section needs a reference. Were the papers assessed on quality, or only on bias risk ? If not, why not, and is this not a shortage ? Have citations or references be screened (‘snowball’). Related to the risk bias measurement, figure 4 is suggested, but is not in the paper.

[ANSWER] The following paragraph has been improved with references

2.4. Risk of Bias measurement

The risk of bias assessment was conducted following the OHAT risk of bias guidelines for humans and animals studies [61]. The risk of bias assessment was performed in accordance to the criteria: confounding bias, detection bias, confidence in the outcome, reporting bias, and other bias. The risk of bias output was categorized as adequate, unclear or inadequate. Then, the papers were categorized as low risk of bias if a minimum ratio of 5/7 positive parameters were detected. On other hands, the articles were considered as high risk of bias studies.

[ANSWER] The fig.4 has been added.

It looks like the discussion is a mix of results and discussion

  1. What is the relevance of 4.1. as these studies read as observational, not so much ‘preventive’ ?

In addition to the observational studies in this subsection 4.1, there are also prospective studies and case reports. These studies were considered so that preventive strategies could be better studied.

  1. The link with folic acid is an association, so not necessary (and quite likely) not causal, but association ?

Inadequate intake of folic acid during pregnancy is a factor risk for skull deformation

  1. What is the relevance of 4.3 if your studies has its focus ‘only’ on prevention, as this describes interventional studies in diagnosed infants (‘therapies’)       

Dear reviewer, as you suggested due to the presence of the sub-section on therapies I have changed the title in “A systematic reviewon preventive practices and therapies for positional plagiocephaly in the developing patient.” to make the text more consistent with the heading.

  1. The 4.4. is disproportional large, and seems to be supported by one study ? and mix of results and discussion

Dear reviewer, we have shortened under your suggestion the sub-paragraph. As the topic is new, there are still few supporting articles extrapolated from the PRISMA analysis, which is why only one study is cited. However, as the study tends to have a focus on prevention, we thought it appropriate to expand on this in a dedicated sub-section.

In conclusion, I value the effort made, but the authors have to extensively reconsider their aims, and the subsequent text structure.

Reviewer 2 Report

Dear Authors,

All the manuscript has gaps in structure and English edition. The authors try to understand the skull's normal morphogenesis and the histological bone tissue proprieties. The authors should improve the technical language in the scientific scope, such as "plastic "skull (line 79). 

The search strategy needs to adjust to the aim of the manuscript. The authors identified two keywords: Positional and Plagiocephaly, but there are deeply correlated

https://meshb.nlm.nih.gov/search?searchMethod=FullWord&searchInField=termDescriptor&sort=&size=20&searchType=allWords&from=0&q=plagiocephaly . This keyword selection cannot be for the systematic review purpose, and it is

not following the title of this paper.  The manuscript assessment, inclusion, and exclusion criteria are not following the clinical issue. Criterium 1) "babies and children aged 0 to 2 years" is not the

correct age range, and it is not under the guidelines of the World Health

Organization. Criterium 2) is missing. Criterium 3) "preventative

interventions" should be clarified; Did the authors intend to search for a

specific healthcare procedure? How can the authors consider the age range 0-2

years old for a clinical follow-up in the scope of the manuscript? The data

processing should be identified regarding adequate guidelines. 

For the above, it is challenging to understand this systematic review. I suggest the authors restructure the research strategy. And also, to make their images! 

Author Response

Dear Authors,

All the manuscript has gaps in structure and English edition. The authors try to understand the skull's normal morphogenesis and the histological bone tissue proprieties. The authors should improve the technical language in the scientific scope, such as "plastic "skull (line 79). 

I corrected it.

The search strategy needs to adjust to the aim of the manuscript. The authors identified two keywords: Positional and Plagiocephaly, but there are deeply correlated

https://meshb.nlm.nih.gov/search?searchMethod=FullWord&searchInField=termDescriptor&sort=&size=20&searchType=allWords&from=0&q=plagiocephaly . This keyword selection cannot be for the systematic review purpose, and it is

not following the title of this paper.  

In this systematic review, we have focused on preventive and therapeutic aspects for positional plagiocephaly. The terms are very closely related but it was necessary to use this research methodology to exclude from it all forms of synostotic plagiocephaly.

The manuscript assessment, inclusion, and exclusion criteria are not following the clinical issue. Criterium 1) "babies and children aged 0 to 2 years" is not the

correct age range, and it is not under the guidelines of the World Health

Organization. Criterium 2) is missing.

Done. It’s added.

Criterium 3) "preventative

interventions" should be clarified;

We clarified it including the use of a hard-surfaced mattress, lateralisation of infant during sleep, and tummy time activities.

Did the authors intend to search for a

specific healthcare procedure?

No, we have evaluated the various possibilities of preventive and therapeutic interventions in the literature without focusing on one specific procedure.

How can the authors consider the age range 0-2

years old for a clinical follow-up in the scope of the manuscript? The data

processing should be identified regarding adequate guidelines.

We used this age range because we were also assessing therapeutic response as well as preventive measures. Since some articles mention forms of positional plagiocephaly in newborn we considered also the first mounth of life.

For the above, it is challenging to understand this systematic review. I suggest the authors restructure the research strategy. And also, to make their images! 

Reviewer 3 Report

This manuscript describes the protocol and findings for a systematic review of the literature around positional plagiocephaly and potential treatment approaches. The search strategy is simple and well described. The approach appears to adhere well to PRISMA guidelines, and the protocol was registered prior to publication. The text itself is disorganized and difficult to read, being mostly made up of single statements that are loosely grouped together. There are places that need clarification. The analysis of findings is weak, and the conclusions are not well supported by the data presented.

- The abstract states that treating plagiocephaly is important for avoiding “functional consequences.” What are functional consequences of plagiocephaly? This would be something to add to the introduction.

- Introduction, line 134-135: This would benefit from more explanation of the disadvantages of helmet therapy. What does “selective efficacy” mean? Helmet therapy may be relatively expensive, but compared to what? It would be hard to believe that it is more expensive than a course of PT, for example.

- Figure 2: I suggest giving more information about the classification of plagiocephaly (ie a textual description of the differences between the 5 types of plagiocephaly). On its own, this figure is not likely to help a reader with no familiarity with the Argenta classification understand the types.

- Figure 3: What is meant by the “Reports not received (n=60) section? Why were they not received? This needs to be explained better.

- Figure 4 appears to be missing.

- In the tables there are a number of abbreviations that are not defined, except in the abbreviation list. It would be helpful to have captions for the tables that define these abbreviations.

- Lines 197-198: This sentence appears to be out of place. Also, this sentence’s connection to Figure 10, which it cites, is not clear.

- Lines 217-223: this section on folate supplementation is confusing, and does not make it clear whether folate helps or hinders with respect to plagiocephaly incidence.

- Table 4, line with Wendling-Keim reference, results column: Are the numbers for CI correct? It appears from what is recorded in the table that PT intervention had a higher remodeling rate, not lower.

- Section 4.4 (about the Welcome Pad) reads like an advertisement, and is not supported by the remainder of the article. It is also not the subject of any of the research reviewed in the systematic review. This section should be removed.

- Line 393: What is “the device” that is high-performing?

- In the discussion of helmet therapy, there is not much of an attempt at synthesizing the findings of HT studies, even though more reviewed papers discuss helmet therapy than other interventions. This would be a helpful addition to the manuscript. It is likewise a glaring omission that helmet therapy is not mentioned in the conclusion to the manuscript.

- In the conclusion (lines 401-404), the authors assert that plagiocephaly can lead to long-term “psychological problems and social discomfort.” However, these concepts were not addressed in the body of the manuscript, and as such do not belong in the conclusion. Either this statement should be removed from the conclusion, or some discussion of these potential sequelae of plagiocephaly should be added to the manuscript.

- Also in the conclusion, the authors assert that orthotic mattresses have “numerous advantages” over other approaches. This conclusion is not really justified by the data presented. There was only a single reference to any orthotic mattress, and this was a relatively small case series (Sillifant et al).

Author Response

This manuscript describes the protocol and findings for a systematic review of the literature around positional plagiocephaly and potential treatment approaches. The search strategy is simple and well described. The approach appears to adhere well to PRISMA guidelines, and the protocol was registered prior to publication. The text itself is disorganized and difficult to read, being mostly made up of single statements that are loosely grouped together. There are places that need clarification. The analysis of findings is weak, and the conclusions are not well supported by the data presented. 

- The abstract states that treating plagiocephaly is important for avoiding “functional consequences.” What are functional consequences of plagiocephaly? This would be something to add to the introduction.

Done

- Introduction, line 134-135: This would benefit from more explanation of the disadvantages of helmet therapy. What does “selective efficacy” mean? Helmet therapy may be relatively expensive, but compared to what? It would be hard to believe that it is more expensive than a course of PT, for example. 

Done. We have redefined the disadvantages.

- Figure 2: I suggest giving more information about the classification of plagiocephaly (ie a textual description of the differences between the 5 types of plagiocephaly). On its own, this figure is not likely to help a reader with no familiarity with the Argenta classification understand the types. 

Done. We added it.

- Figure 3: What is meant by the “Reports not received (n=60) section? Why were they not received? This needs to be explained better. 

The section of “Reports not retrieved” represents the number of articles for which you are unable to find the full text, both by contacting various authors but also searching in Find@UNC and Interlibrary Loan.

- Figure 4 appears to be missing. 

There is the figure at line 206 with description of the risk of bias.

- In the tables there are a number of abbreviations that are not defined, except in the abbreviation list. It would be helpful to have captions for the tables that define these abbreviations. 

Done. We defined them.

- Lines 197-198: This sentence appears to be out of place. Also, this sentence’s connection to Figure 10, which it cites, is not clear. 

Done, it’s corrected. The sentence, with the right correction to figure 10, is in lines 403-405, with a logical connection with the speech preceding and following it.

- Lines 217-223: this section on folate supplementation is confusing, and does not make it clear whether folate helps or hinders with respect to plagiocephaly incidence. 

Done. Folic acid helps prevent neural tube defects as we pointed out in line 258.

- Table 4, line with Wendling-Keim reference, results column: Are the numbers for CI correct? It appears from what is recorded in the table that PT intervention had a higher remodeling rate, not lower. 

Yes. It’s correct.

- Section 4.4 (about the Welcome Pad) reads like an advertisement, and is not supported by the remainder of the article. It is also not the subject of any of the research reviewed in the systematic review. This section should be removed.

We have evaluated and verified in our opinion that the device performs better than the others compared. This is not an advertisement but a comparison of existing medical devices on the market in order to assess the best therapeutic response for these patients. This device is new, which is why it is not supported by a large amount of articles.

- Line 393: What is “the device” that is high-performing? 

We are referring to mattress.

- In the discussion of helmet therapy, there is not much of an attempt at synthesizing the findings of HT studies, even though more reviewed papers discuss helmet therapy than other interventions. This would be a helpful addition to the manuscript. It is likewise a glaring omission that helmet therapy is not mentioned in the conclusion to the manuscript. 

Done. Discussion and conclusion have been improved.

- In the conclusion (lines 401-404), the authors assert that plagiocephaly can lead to long-term “psychological problems and social discomfort.” However, these concepts were not addressed in the body of the manuscript, and as such do not belong in the conclusion. Either this statement should be removed from the conclusion, or some discussion of these potential sequelae of plagiocephaly should be added to the manuscript. 

Done. We have removed that.

- Also in the conclusion, the authors assert that orthotic mattresses have “numerous advantages” over other approaches. This conclusion is not really justified by the data presented. There was only a single reference to any orthotic mattress, and this was a relatively small case series (Sillifant et al).

It is a new device, which is why we found only one valid article in the literature to support the search criteria.

Reviewer 4 Report

satisfactory

Author Response

Thank you very much.

Reviewer 5 Report

Comments are attached 

Author Response

Copy and paste from different sources without showing any sort of clinical experience in dealing

with plagiocephaly is not a good approach. Examining a child with plagiocephaly is a heavy

responsibility for pediatricians. Dozens of children received misdiagnosis and were victims of

false interpretations of plagiocephaly. The latter is an extremely sensitive and difficult area and

can never jump over through rash and inaccurate assumptions OR treatment. Prompt clinical

experience is the corner stone in the management of cranial deformation. The sequence of order

of the management should include the followings: Firstly, pediatricians /physicians need to

examine the child in full details, parents should be included, clinical observations are mandatory

for any unusual craniofacial/musculoskeletal abnormalities regardless the severity. Secondly,

clinical examination of the siblings along with gathering of information for the rest of the family

subjects. Thirdly, pediatricians must exclude all forms of abnormal craniofacial contour and its

allied syndromic associations. In the text books of syndromic associations and skeletal dysplasia,

there are more than 72 heterogeneous group of syndromic entities which meet the searching

criteria of plagiocephaly. In addition to non-instantly recognizable and confusing forms of

skeletal sclerosing disorders in which craniotubular deformities encompassing a long list. Finally,

I would like to say in accordance with long term experience with plagiocephaly and skull

deformational pathologies. That, the following statement by the authors of connecting the

increased risk of malformation of the skull to inadequate vitamin D consumption in early infancy

and during the final trimester of pregnancy, is unrealistic assumption.

Authors need to dig deeper in clinical medicine and follow the above mentioned points.

Answer

We have incorporated your suggestions in the introduction even though our work is based on positional plagiocephaly.

Round 2

Reviewer 1 Report

My main concern relates to the fact that the authors have extended their search strategy (at least in the title and abstract), but without sufficient considering the physiotherapy. This is a concern, that can in my opinion be addressed by explicit mentioning that physiotherapy was not considered by only 'other' interventions ? 

Author Response

REVIEWER 1 REPORT ROUND 2 

My main concern relates to the fact that the authors have extended their search strategy (at least in the title and abstract), but without sufficient considering the physiotherapy. This is a concern, that can in my opinion be addressed by explicit mentioning that physiotherapy was not considered by only 'other' interventions ? 

We focused on the use of preventive and therapeutic devices rather than on operator manipulation.

Reviewer 2 Report

Dear Authors,

The authors did not correspond to all the suggestions performed in the previous revision. I suggest reading carefully and improving the highlights concerning the material and method section. The authors said, "The terms are very closely related, but it was necessary to use this research methodology to exclude all forms of synostotic plagiocephaly,” so it should be added to the research strategy. Do the figures have legal consent for their use? I suggest that the authors should make new figures.

Author Response

REVIEWER 2 REPORT ROUND 2     

Dear Authors,

The authors did not correspond to all the suggestions performed in the previous revision. I suggest reading carefully and improving the highlights concerning the material and method section. The authors said, "The terms are very closely related, but it was necessary to use this research methodology to exclude all forms of synostotic plagiocephaly,” so it should be added to the research strategy. Do the figures have legal consent for their use? I suggest that the authors should make new figures.

Dear Reviewer, as we were exposed to you  in the previous report, it is not a goal of our research strategy to include all forms of synostotic plagiocephaly, which were only mentioned for rigor of form in the introduction, and therefore we do not want to do a systematic review on synostotic and nonsynostotic plagiocephaly. Our aim was to investigate about positional plagiocephaly. Figures have been removed or modified.

Reviewer 3 Report

The authors have addressed most of my concerns. I still find it unusual to spend so much of the text of a systematic review paper on a specific commercial product that is not evaluated in any of the studies reviewed. Although the authors noted in their response that this product is highlighted because of their experience using it, it is still odd to go through the work of a systematic review then conclude with highlighting something other than what was found in the review. The rationale for why this is settled on rather than interventions with much more research behind them is not particularly developed in the text, and the section about the Welcome Pad does not make it clear that this is due to the authors’ own experience. This section also highlights information about the Welcome Pad which is not explained and/or not supported by cited literature. For example, Figure 6 highlights “Thermoregulation effect” which is not explained. Right after this, the term “Hygiene Plus” is used, but not defined. This term and others (such as “sleep secure,” “density secure,” and “supershape” are not standard academic or clinical terms, and rather sound like marketing terms. Also, the idea of improving tactile sensitivity/motor skills because of the folds on the mattress is not supported by any literature, and thus is another claim about this item that should not be made in an academic paper.  I would recommend removing most of this section for the paper. If the authors are determined to keep it, it really needs to be made much more clear that this is due to their personal experience and that they recommend this as clinicians, because this is not justified by the systematic review that makes up the rest of the paper.  

Author Response

REVIEWER 3 REPORT ROUND 2                                                 

The authors have addressed most of my concerns. I still find it unusual to spend so much of the text of a systematic review paper on a specific commercial product that is not evaluated in any of the studies reviewed. Although the authors noted in their response that this product is highlighted because of their experience using it, it is still odd to go through the work of a systematic review then conclude with highlighting something other than what was found in the review. The rationale for why this is settled on rather than interventions with much more research behind them is not particularly developed in the text, and the section about the Welcome Pad does not make it clear that this is due to the authors’ own experience. This section also highlights information about the Welcome Pad which is not explained and/or not supported by cited literature. For example, Figure 6 highlights “Thermoregulation effect” which is not explained. Right after this, the term “Hygiene Plus” is used, but not defined. This term and others (such as “sleep secure,” “density secure,” and “supershape” are not standard academic or clinical terms, and rather sound like marketing terms. Also, the idea of improving tactile sensitivity/motor skills because of the folds on the mattress is not supported by any literature, and thus is another claim about this item that should not be made in an academic paper.  I would recommend removing most of this section for the paper. If the authors are determined to keep it, it really needs to be made much more clear that this is due to their personal experience and that they recommend this as clinicians, because this is not justified by the systematic review that makes up the rest of the paper. 

The section has been modified. Our study was based on the 'analysis of the characteristics of individual mattresses on the market, unfortunately, there are no data in the literature and only one existing article is cited (95).

The focus on this mattress came from the presence of better-performing characteristics, for the specific period of infant growth (0 to 6 months).

In addition, it is an evolution of a device considered in an experimental study, by professionals from the departments of Neonatology and Neonatal Intensive Care at the Ospedale Maggiore in Bologna and at the Infant Rehabilitation Medicine at Bellaria, of which no data have been published.

Reviewer 5 Report

Minor revision is needed. Instructions are attached 

Author Response

REVIEWER 5 REPORT ROUND 2                

Minor revision is needed. Instructions are attached 

Line 249: An increased skull malformation risk has been linked to inadequate vitamin D consumption in early infancy and during the final trimester of pregnancy.

This statement or study is just assumptive and can never considered as a source of precious

clinical information. Delete reference 71

Dear reviewer thank you for the suggestion, but we considered this work because it is an observational case–control study, suggesting that the reduced intake of vitamin D in early infancy and during the final trimester of pregnancy is associated with a higher risk of positional skull deformation .

Line 116-line 135 ( types of plagiocephaly)

Authors should add, if plagiocephaly is associated with any sort of abnormal facial features such as bulging eyes, abnormal positions of the ears, abnormal zygomatic or mandibular bones are or facial asymmetry and so forth. Pediatricians should put in mind the diagnosis of craniosynostosis, until prove otherwise.

The various degrees of clinical manifestations of plagiocephaly have been described in detail in the classification.

References

Only two references re-lambdoid synostosis insufficient.

Add more studies for syndromic and non-syndromic craniosynostosis (more references are

required).

Some references have been added (5,9,10).

Lessens the huge number of the references

Eight references have been removed.

Round 3

Reviewer 1 Report

no additional comments 

Author Response

Thank you very much!

Reviewer 2 Report

Dear Authors,

The design of a systematic review is based on a question and its answer guided by keywords associated with it but not dependent on it. The authors performed a search methodology based on two words considered interrelated or dependent (I suggest https://meshb.nlm.nih.gov/search?searchMethod=FullWord&searchInField=termDescriptor&sort=&size=20&searchType=allWords&q=Positional%20plagiocephaly&from=0)

This reviewer suggested highlighting and excluding all forms of synostotic plagiocephaly. It would be interesting to add the word "prevention" (https://meshb.nlm.nih.gov/search?searchMethod=FullWord&searchInField=termDescriptor&sort=&size=20&searchType=allWords&from=0&q=prevention). It will align with the study's objective and exclude synostotic plagiocephaly for its therapeutic approach.
This suggestion is the core of the work carried out and leads to structural changes, but from my point of view, it is necessary!
As for the figures, with copyright protection, I consider a positive answer.

Author Response

Dear Reviewer, although 'prevention' was not included as a keyword, it was considered in the search strategy at the screening stage.

Reviewer 3 Report

The authors addressed some of my remaining concerns, but not others. Specifically:

- It is still not clear from the manuscript why the authors chose to spend so much of this manuscript talking about the Welcome Pad in particular. In the authors’ response, they state “The focus on this mattress came from the presence of better-performing characteristics…” How is this concluded, though, since there is no literature on this product?

- Similarly, the explanation that the Welcome Pad is “an evolution of a device considered in an experimental study…” also does not make it clear why it is being afforded so much space here. The previous device also does not appear to have any published studies to support its use (and appears to have been developed for prevention of SIDS/SUID rather than for positional plagiocephaly). The citation added for this is for a news release rather than for a research study. Thus, it is not clear why this is a device “considered in an experimental study.”

- The revisions do not make it clear that the section about the Welcome Pad is not a conclusion based on the systematic review. For example, the first sentence of this section (line 329) states that “Current management of PP involves the use of orthotic mattress.” This implies that orthopedic mattress use is a standard of care. However, given the paucity of literature on orthotic mattresses, this really isn’t justified in a systematic review. This whole section still seems out of place in this type of paper, but at the least the authors should start this section out making it clear that this is personal practice experience. For example, they could start this section along the lines of “One of the options for treatment/prevention of PP involves the use of orthotic mattresses. Although there is not yet much published evidence to support this approach, in our experience …” Similarly, when the Welcome Pad itself is introduced (line 350), the authors do state “there is no evidence in the literature,” but then go on to describe it in language that implies an evidence base (e.g. line 351, “was found to present more features to enable comfort, hygiene, and safety…”).

- The paragraph on the lower part of the Welcome Pad (lines 414-418) describes the “deep welds” in the mattress and asserts that they improve motor skills, but do not cite any research to support this assertion. Since there are no publications using this mattress, it really would be best to leave this section out. This was noted in a previous review, but was not addressed by the authors in this revision.

Author Response

Dear Reviewer, there is only one article to support it because it is an innovative device. We want to help patients by using this innovative pad to support the prevention of positional plagiocephaly. We have made it clear that the section on the welcome pad is based on our personal experience and is therefore not the result of the literature review. We have removed another section you suggested.